# Auxiliary Variational MCMC

**Raza. Habib**
Department of Computer Science
University College London
raza.habib@cs.ucl.ac.uk

**David. Barber**
Department of Computer Science
University College London
david.barber@ucl.ac.uk

## Abstract

We introduce Auxiliary Variational MCMC, a novel framework for learning MCMC kernels that combines recent advances in variational inference with insights drawn from traditional auxiliary variable MCMC methods such as Hamiltonian Monte Carlo. Our framework exploits low dimensional structure in the target distribution in order to learn a more efficient MCMC sampler. The resulting sampler is able to suppress random walk behaviour and mix between modes efficiently, without the need to compute gradients of the target distribution. We test our sampler on a number of challenging distributions, where the underlying structure is known, and on the task of posterior sampling in Bayesian logistic regression. Code to reproduce all experiments is available at https://github.com/AVMCMC.

## 1    Introduction

Markov Chain Monte Carlo (MCMC) and Variational Inference (VI) are well-established approaches to approximating expectations under the complex distributions $p$ that frequently arise in machine learning and statistics (Wainwright & Jordan, 2008; Brooks et al., 2011). VI is usually fast and cheap but often places strong restrictions on the class of approximating distributions leading to irreducible bias. MCMC on the other hand is asymptotically exact but may require an infeasible amount of computation to converge. Given the complimentary strengths and weaknesses of the two methods it is natural to wish to combine them (De Freitas et al., 2001; Salimans et al., 2015).

The most naive pairing of the two methods, simply using a variational approximation $q$ as the proposal distribution in a Metropolis-Hastings sampler (see for example Gamerman & Lopes (2006) or De Freitas et al. (2001)), is known to scale poorly with the dimension of the target distribution (De Freitas et al., 2001) and mix inefficiently when $p$ is multi-modal.

In this paper we suggest an alternative approach inspired both by the successes of black box variational inference (Ranganath et al., 2014) and auxiliary variable MCMC methods, such as Hamiltonian Monte Carlo (HMC) (Duane et al., 1987; Girolami & Calderhead, 2011).

The key contributions of our work are :

- A general framework for marrying variational inference with Markov Chain Monte Carlo in a way likely to produce efficient samplers.
- The use of the auxiliary variational method to capture latent low-dimensional structure in our target distributions and exploit this structure to suppress random walk behaviour.
- The extension of the Metropolis-Hastings algorithm to continuous mixture proposals.
- The introduction and demonstration of a specific instance of our framework, the Auxiliary Variational Sampler (AVS). Our sampler takes advantage of flexible distributions parameterized by neural networks that can be trained in a fully black-box manner.

## 2    Auxiliary variational MCMC

The key idea behind Auxiliary Variational MCMC is to exploit structure present in the target distribution $p(x)$ by first fitting a parameterized variational approximation in an augmented space. This allows the sampler to leverage learned low-dimensional structure. In the subsequent sections

we introduce the auxiliary variational method and describe how it can be combined with a carefully chosen class of proposal distributions to construct an efficient sampler.

## 2.1 MIXTURE PROPOSAL MCMC

To develop a valid MCMC algorithm we need to construct an ergodic Markov chain whose stationary distribution is our target distribution $p(x)$. In order to do this we introduce a Metropolis-Hastings (Gamerman & Lopes, 2006) like algorithm with a specially chosen form of proposal distribution. This proposal can be naturally combined with the auxiliary variational method which we will introduce in section 2.2. We first consider a mixture proposal distribution of the following form[1]:

$$\tilde{q}(x'|x) = \int \tilde{q}(x'|a)\tilde{q}(a|x)da \tag{1}$$

We prove in supplement (7) that the following forms a valid MCMC sampling step:

1. Sample $a$ from $\tilde{q}(a|x)$
2. Sample $x'$ from $\tilde{q}(x'|a)$
3. Accept the candidate sample $x'$ with probability

$$\min\left\{1, \frac{\tilde{q}(x|a)\tilde{q}(a|x')p(x')}{\tilde{q}(x'|a)\tilde{q}(a|x)p(x)}\right\} \tag{2}$$

otherwise reject $x'$ and define the new sample as a copy of the current $x$, namely $x' = x$.

It is worth noting that the above procedure is not equivalent to simply performing Metropolis-Hastings in the joint $(x, a)$ space, but is a sampler in $x$ alone, specified by the marginalized proposal distribution given by (1). The two become equivalent only in the case that $\tilde{q}(a|x)$ is constant, see supplement (7).

The mixture proposal can be extended to an arbitrary number of auxiliary variables as long as the acceptance ratio is adjusted accordingly. For example with two auxiliary variables, the proposal becomes:

$$\tilde{q}(x'|x) = \int \tilde{q}(x'|a')\tilde{q}(a'|a)\tilde{q}(a|x)dada' \tag{3}$$

where $\tilde{q}(a'|a)$ is another arbitrary proposal distribution. The acceptance probability is now given by:

$$\min\left\{1, \frac{p(x')\tilde{q}(x|a')\tilde{q}(a'|a)\tilde{q}(a|x')}{p(x)\tilde{q}(x'|a)\tilde{q}(a|a')\tilde{q}(a'|x)}\right\} \tag{4}$$

For a proposal $\tilde{q}(a'|a)$ that is symmetric, $\tilde{q}(a'|a) = \tilde{q}(a|a')$, this simplifies to:

$$\min\left\{1, \frac{p(x')\tilde{q}(x|a')\tilde{q}(a|x')}{p(x)\tilde{q}(x'|a)\tilde{q}(a'|x)}\right\} \tag{5}$$

We now show how the mixture proposal can be naturally combined with the auxiliary variational method.

## 2.2 THE AUXILIARY VARIATIONAL METHOD

The auxiliary variational method (Agakov & Barber, 2004; Ranganath et al., 2016) is a strategy for creating more expressive families of approximating distributions for use in variational inference. Instead of minimizing the Kullback-Leibler divergence between our approximating distribution $q(x)$ and our target $p(x)$, we instead minimize the divergence in an augmented space $(x, a)$ with additional auxiliary variables $a$. In doing this, we first define a joint $q_\phi(x, a)$ in the augmented space and a joint $p(x, a) = p_\theta(a|x)p(x)$ where $\theta$ and $\phi$ are parameters. Marginalizing $p(x, a)$ over $a$ recovers $p(x)$ by construction. We are free to decide on the dimension and form (e.g. continuous, discrete or mixed) for $a$ and our objective is the Kullback-Leibler divergence between the joint approximation $q$ and joint target $p$:

$$(\phi^*, \theta^*) = \underset{\phi, \theta}{\arg\min} \, \mathrm{KL}\left(q_\phi(x, a)||p(x)p_\theta(a|x)\right) \tag{6}$$

---

[1]In general we use $\tilde{q}$ to denote a proposal distribution and $q$ to denote a variational distribution.

---

**Algorithm 1:** Auxiliary variational sampler

---

Initialize $\phi$ and $\theta$     ▷ Fit the variational distribution
**while** *Not Converged* **do**

  $\{a_1, \ldots, a_N\} \sim q_\phi(a)$
  $\{x_1, \ldots, x_N\} \sim q_\phi(x|a_n)$      ▷ re-parameterized
  $L \leftarrow \frac{1}{N} \sum_n \log \frac{q_\phi(x_n|a_n)q_\phi(a_n)}{p(x_n)p_\theta(a_n|x_n)}$
  $\phi \leftarrow \phi - \eta \nabla_\phi L$
  $\theta \leftarrow \theta - \eta \nabla_\theta L$

**end**
$a \sim q_\phi(a)$               ▷ Mixture-Model MCMC sampling
$x_0 \sim q_\phi(x|a)$
**for** $t = 0, \ldots, T$ **do**

  $a \sim p_\theta(a|x_t)$
  $a' \sim \mathcal{N}\left(a'|a, \sigma_a^2 I\right)$
  $x' \sim q_\phi(x|a')$
  $A \leftarrow \frac{p(x')p_\theta(a'|x')q_\phi(x_t|a)}{p(x_t)p_\theta(a|x_t)q_\phi(x'|a')}$      ▷ acceptance ratio
  $u \sim U[0,1]$ ▷ sample $u$ from uniform distribution
  **if** $u \leq A$ **then**
  |   $x_{t+1} = x'$
  **else**
  |   $x_{t+1} = x_t$
  **end**

**end**

---

The Auxiliary Variational Sampler. The algorithm begins by fitting a mixture variational distribution to the target $p(x)$ by stochastic gradient descent, based on the auxiliary variational method.

After the auxiliary variational model is fitted to $p(x)$ the MCMC sampling phase begins and is a modified form of Metropolis-Hastings sampling.

Moving to the joint space allows us to tractably learn complex approximating distributions $q_\phi(x, a)$ whose marginal, $q_\phi(x) = \int q_\phi(x|a)q_\phi(a)da$, may be intractable.

Whilst sampling in high-dimensional spaces is difficult, in many cases of interest the high probability or typical regions lie close to much lower dimensional manifolds. A key point of fitting an auxiliary variational distribution is that, by constraining $a$ to have dimension much lower than $x$, we can then exploit this low dimensional manifold to form a more efficient proposal distribution, staying close to the manifold of significant probability.

### 2.3 COMBINING AUXILIARY VARIATIONAL INFERENCE AND MCMC

After fitting our variational approximation to $p(x)$, we will have three variational distributions: $q_*(x|a)$, $q_*(a)$ and $p_*(a|x)$ that are the solution of the optimization problem given in (6). These distributions approximately satisfy the following relationships:

$$\int p(x)p_*(a|x)dx \approx q_*(a), \qquad \int q_*(x|a)q_*(a)da \approx p(x) \tag{7}$$

which become exact iff the divergence in (6) becomes zero. Here $p_*(a|x)$ is a learned stochastic mapping from the high-dimensional target space to the low-dimensional auxiliary space and $q_*(x|a)$ is a mapping in the opposite direction. These learned mappings can be composed in a variety of ways to form marginal proposal distributions of the kind discussed in section 2.1 and also joint proposals. We now discuss some natural combinations and argue that these may form good proposal distributions in practice.

#### 2.3.1 NAIVE PROPOSAL DISTRIBUTIONS

A simple proposal constructed from our variational distribution is to perform Metropolis-Hastings in the joint $(x, a)$ space with an independent proposal given by:

$$\tilde{q}(x', a'|x, a) = q_*(x'|a')q_*(a') \tag{8}$$

If the variational approximation is accurate such that $q(x, a) \approx p(x)p(a|x)$ then one might expect this to have high acceptance probability. A potential downside of this scheme is that proposals are independent between time-steps and in high dimensions this may cause the acceptance ratio to become impractically low.

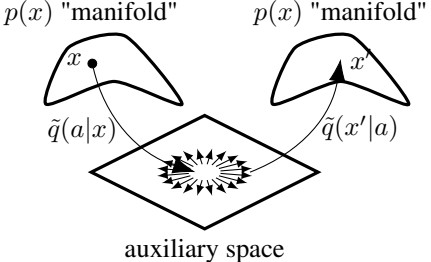

Figure 1: The auxiliary random walk proposal. The variational approximation allows for small steps in the auxiliary space to correspond to large steps in the target space. Our initial point starts on a manifold of high probability in the target space, is mapped down to the low-dimension auxiliary space, perturbed, and then mapped back up to the high probability manifold. Unlike a random-walk in the $x$-space, our random perturbations correspond to moves along the high probability manifold.

Another natural MCMC proposal distribution to consider is:

$$\tilde{q}(x'|x) = \int q_*(x'|a)p_*(a|x)da \qquad (9)$$

where we have replaced the arbitrary proposal distribution of (1) with the optimal variational distributions learned by minimizing the joint divergence given in (6). We might expect this to be a promising proposal distribution, with high acceptance probability, as it already approximately satisfies the stationarity criterion of our MCMC chain. That is

$$\int \tilde{q}(x'|x)p(x)dx = \int q_*(x'|a)p_*(a|x)p(x)dxda \approx \int q_*(x'|a)q_*(x|a)q_*(a)dxda$$

$$= \int q_*(x'|a)q_*(a)da \approx \int p_*(a|x')p_*(a)da = p(x')$$

However, if the variational distribution truly captures underlying structure in $p(x)$, it has been our experience that $p_*(a|x)$ and $q_*(x|a)$ become approximate inverses causing $\tilde{q}(x'|x)$ to resemble the identity mapping and thereby slowing mixing in the chain.

### 2.3.2 AN AUXILIARY RANDOM WALK PROPOSAL DISTRIBUTION

To avoid the above identity mapping issue and encourage the sampler to take large steps, we introduce an additional random perturbation in the auxiliary $a$-space. That is we use a proposal distribution:

$$\tilde{q}(x'|x) = \int q_*(x'|a')\tilde{q}(a'|a)p_*(a|x)dada' \qquad (10)$$

where $\tilde{q}(a'|a) = \mathcal{N}\left(a'|a, \sigma_a^2 I\right)$ is a Gaussian proposal with mean $a$ and isotropic covariance $\sigma_a^2 I$. The overall algorithm can be viewed as mapping from the high-dimensional $x$ to the low-dimensional $a$, performing a random walk in the low dimensional auxiliary $a$-space and subsequently mapping back up to the high-dimensional target space $x$, see figure (1).

In high dimensions our target distribution is likely to have high probability only close to some low-dimensional manifold. The traditional Random-Walk-Metropolis Algorithm proposes new samples $x'$ by perturbing the most recent sample in an arbitrary direction, usually based on a Gaussian proposal $\tilde{q}(x'|x) = \mathcal{N}\left(x'|x, \sigma_x^2 I\right)$; however, in high-dimensional spaces almost all directions will correspond to steps off the manifold of high probability and thus out of the typical set. In our case, we perform the random perturbation in the low-dimensional auxiliary space and our variational distribution $q_*(x'|a')$ ensures that this move remains within the manifold of high density.

We depict this process in figure (1) and provide explicit examples of the learned latent structure of real distributions in section 3. Much like HMC, we are able to exploit the addition of auxiliary variables to encourage large moves of high probability in the target space but unlike HMC we do so by explicitly modelling the structure of the target distribution and do not require gradients of the target density. We are able to take large steps because small perturbations in $a$ can correspond to

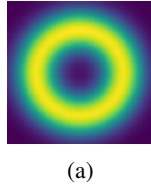

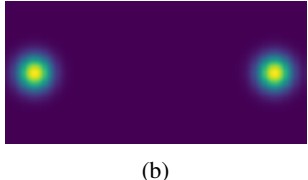

(a)                                                        (b)

Figure 2: Target densities with a high degree of latent structure. (a) A ring of high density centered at the origin. (b) Mixture of Two Gaussians with highly separated means.

large perturbations *within the manifold of high probability*; unlike methods that adapt to the local geometry of the probability manifold (Girolami & Calderhead, 2011; Strathmann et al., 2015), our variational fit allows for larger, non-local moves in the $x$-space. Note that, unlike HMC, in general there is no requirement that $(x, a)$ are continuous random variables.

## 2.4 Choosing the variational family

Within the above framework, we still have to decide on the structure of the variational distributions, $q(a, x)$ and $p(a|x)$. We take inspiration from recent successes in the generative modeling of complex data distributions (Kingma & Welling, 2014), and propose to parameterize our variational distributions using deep neural networks. For continuous $x$ we choose the following structure for each of our approximating distributions:

$$q(a) = \mathcal{N}(a|0, I) \tag{11}$$
$$q_\phi(x|a) = \mathcal{N}(x|\mu_\phi(a), \Sigma_\phi(a)) \tag{12}$$
$$p_\theta(a|x) = \mathcal{N}(a|\mu_\theta(x), \Sigma_\theta(x)) \tag{13}$$

Where $q_\phi(x|a)$ and $p_\theta(a|x)$ are both diagonal Gaussian distributions whose means and covariances are parameterized by neural networks with parameters shared between the mean and covariance. In some experiments we also choose $p_\theta(a|x)$ to be a mixture of diagonal Gaussians (see the supplementary material). Key to the flexibility of the auxiliary variational method is that whilst $q(a, x)$ can be evaluated point-wise, the marginal $q(x)$ is a much richer approximating distribution whose density we typically cannot evaluate point-wise. Whilst we can thus evaluate our joint approximating density we still can't compute the objective:

$$\text{KL}(q_\phi(x|a)q(a)||p_\theta(a|x)p(x)) \tag{14}$$

However, we recognize that an unbiased estimator of the KL divergence can be obtained by sampling from $q_\phi(x, a)$. We use the standard re-parameterization trick of Kingma & Welling (2014) to reduce the variance in the corresponding gradient estimator. We refer to this version of the algorithm as the Auxiliary Variational Sampler (AVS). The full algorithm is given in algorithm (1). Re-parameterisation requires the gradient of $\log p(x)$; if this is prohibitively expensive then the score-function estimator may be used (Kleijnen & Rubinstein, 1996). Within this framework, different mixture proposals could be considered, based on the fitted variational distribution, but our experience is that the Auxiliary Random Walk proposal is effective in our experiments.

## 3 Experiments

To demonstrate the benefit of fitting an auxiliary variational approximation we first test our sampler on a number of distributions with known low-dimensional structure. We show how AVS is both able to recover the latent structure and exploit it for the purposes of sampling. We then demonstrate the sampler on more realistic problems.

**Ring density** The first distribution is in two dimensions $x = (x_1, x_2)$. The distribution is a ring of high probability, centered at a fixed distance from the origin, figure (2b). We construct the ring density to have explicit latent structure by defining the target distribution as:

$$p(x) \propto \int_0^{2\pi} e^{-\frac{1}{2\sigma^2}(x - r\mu(\theta))^2} d\theta, \qquad \mu(\theta) = (\cos(\theta), \sin(\theta)) \tag{15}$$

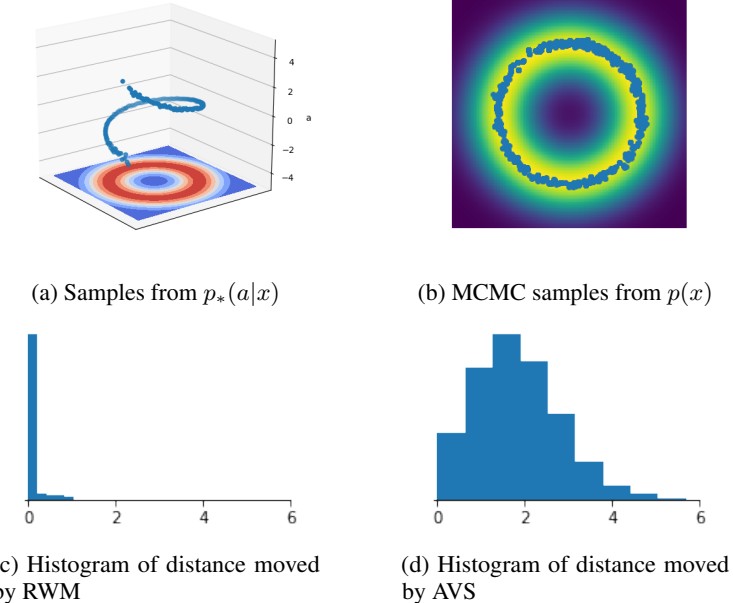

(a) Samples from $p_*(a|x)$

(b) MCMC samples from $p(x)$

(c) Histogram of distance moved by RWM

(d) Histogram of distance moved by AVS

Figure 3: (a) Samples from the learned variational distribution $p_\theta(a|x)$ for $x$ on the ring of high probability. The auxiliary space is plotted on the vertical axis, with the target space plotted in the plane. We can see that the auxiliary variational approximation has recovered the low dimensional structure of the target distribution, so that random perturbations in the auxiliary space will mostly correspond to proposals within the region of high target probability and large moves. (b): Samples from the trained Auxiliary Variational Sampler (AVS) algorithm (1). (c,d) show the distance moved per sampling step for RWM and AVS for a ring of radius 5, thus demonstrating the benefit of exploiting structure.

Since this ring distribution is highly constrained, pure random walk sampling in $x$ will not efficiently move around the ring. However, provided we can find a variational fit to capture the ring structure, we expect our AVS sampler to be able to move more efficiently. To train our AVS sampler, we fit an auxiliary variational approximation with a 1 dimensional continuous mixture (see supplementary for details). We can examine the latent structure learned by the variational distribution by drawing samples from $p_*(a|x)$ as we vary $x = (\cos(\theta), \sin(\theta))$ for $\theta \in [0, 2\pi]$. Looking at figure (3), we can see that the variational distribution has successfully captured the latent structure, automatically learning to map points of high probability in $x$ onto distinct auxiliary variables. Used with the mixture proposal (10) this results in an effective sampler that is able to take much larger step sizes than random walk Metropolis (RWM) as shown in figure (2).

**Mixtures of Gaussians and Student T**    The second pair of densities we wish to sample from is a two dimensional mixture of Gaussians with highly separated means (with a distance of 20 between the means and a standard deviation of 1), figure (2), and a two dimensional mixture of Student T distributions. These distributions have discrete latent structure and are challenging both for random walk Metropolis (RWM) (Gamerman & Lopes, 2006) and more advanced samplers such as HMC, which fail to find more than one mode in any reasonable amount of time or can struggle with the heavy tails. Our sampler has no problem finding both modes and, as shown by the plot of consecutive samples in figure (4), is able to hop between modes in single steps. We use a 1-dimensional continuous auxiliary variable.

**Posterior sampling in Bayesian models**    The low-dimensional distributions above demonstrate the ability of our sampler to learn and exploit latent structure. Here we analyze the performance of our sampler on a more realistic problem: posterior sampling in Bayesian logistic regression. We use the heart data-set used by Song et al. (2017), which has 13 covariates and 270 data-points. We tune all algorithms to maximize effective sample size (see supplement (9)) at convergence. In the case of

(a)

(b)

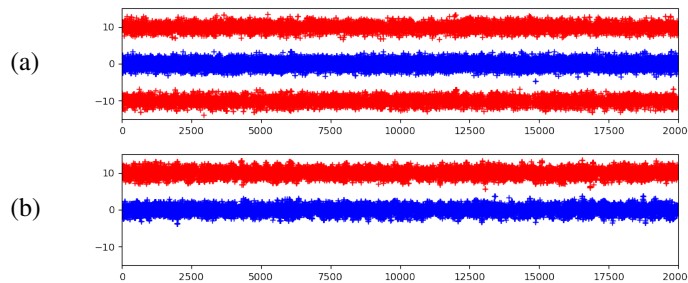

Figure 4: Traceplot showing 20000 consecutive samples of $x_1$ (blue), $x_2$ (red), drawn from the mixture of Gaussians, figure 2, using (a) AVS given in algorithm 1, (b) RWM. AVS can easily move between the modes at $x_1 = 10$ and $x_1 = -10$, whilst RWM get stuck in one mode.

HMC we use a number of initial runs to select an appropriate step-size and then tune the number of leapfrog steps. In all cases, we run a chain for 10000 burn-in steps and then draw 20000 samples to calculate diagnostics. In order to assess convergence and ensure the validity of the results, we monitor the Gelman-Rubin statistic (Gelman & Rubin, 1992) and visually inspect trace-plots of parameters.

### 3.1 EVALUATING PERFORMANCE

Evaluation of MCMC methods is notoriously difficult (Gelman & Rubin, 1992). Even diagnosing convergence is challenging and there is no clear agreement on the "correct" metric of performance. In practice there are two factors that are of primary concern to an end-user: How easy is the sampler to implement and tune? and how computationally efficient is the resulting sampler? We evaluate our performance by comparison to RWM, HMC and two recently proposed neural adaptive samplers: A-NICE-MC (Song et al., 2017) and L2HMC (Levy et al., 2018), discussed further in section 4. For learned samplers there is an additional cost involved in training the sampler that may be offset by improved performance. As a baseline, we include the independent proposal given by equation (8).

We choose here to focus our quantitative evaluation on the *effective Sample Size* (ESS), which measures how many independent samples would be equivalent to an MCMC sample, and the training-time for the learned samplers. We choose not to focus our investigation on burn-in time since AVS, being initialized by the variational distribution, often has short burn-in relative to competitive methods. For ESS calculations, we use the batch-means estimator applied to a single long chain as it has been shown to be more reliable than many alternatives (Thompson, 2010), exact details are given in the supplement. For the experiments with mixtures of Gaussians we do not present ESS for HMC and RWM as both algorithms clearly fail to converge, getting trapped in one of the local modes. For all distributions except the Mixture of Gaussians, the independent sampler also failed, getting quickly stuck in a region of high-probability where the acceptance rate dropped to near zero. To ensure consistency in timing, all methods were implemented using Tensorflow 1.10 (Abadi & Agarwal, 2015) and run on a single Tesla K80 GPU. Code to reproduce all experiments can be found at github.com/AVMCMC. For L2HMC and A-NICE-MCMC, we perform a random search over hyper-parameters ensuring we include the parameters used in the original papers wherever possible. The figures reported are the mean of 10 independent runs alongside their standard deviations.

Table 1: ESS calculated using the batch-means estimator.

|  | AVS | AVS-IND | HMC | RWM | ANICE | L2HMC |
|---|---|---|---|---|---|---|
| Ring | $0.176 \pm 0.005$ | NA | $\mathbf{0.612 \pm 0.120}$ | $0.024 \pm 0.005$ | $0.541 \pm 0.105$ | $0.247 \pm 0.062$ |
| Mix. of Gauss. | $0.178 \pm 0.042$ | $0.009 \pm 0.004$ | NA | NA | $\mathbf{0.322 \pm 0.103}$ | $0.170 \pm 0.10$ |
| Mix. of Student T. | $0.047 \pm 0.026$ | NA | $0.0020 \pm 0.0002$ | $0.002 \pm 0.001$ | $0.071 \pm 0.026$ | NA |
| Log. regression | $0.066 \pm 0.027$ | NA | $0.070 \pm 0.015$ | $0.013 \pm 0.002$ | $0.257 \pm 0.035$ | $\mathbf{0.562 \pm 0.071}$ |

## 4 DISCUSSION AND RELATED WORK

The results demonstrate that Auxiliary MCMC can be used to construct a practical sampler that is able to exploit low dimensional structure and mix between modes. AVS produces competitive ESS, and initializing from a variational approximation reduces the need to burn-in. AVS also offers a straightforward method to interpolate between pure VI and MCMC. We found that the naive baseline, independent Metropolis-Hastings in the joint auxiliary-target space, failed to produce reasonable

Table 2: ESS/s taking into account the total time including both training time and sampling time for 20000 samples (left) and taking account only the sampling time (right). Absolute times are provided in the appendix.

| | AVS | ANICE | L2HMC |
|---|---|---|---|
| Ring | **5.30e-3** | 1.25e-3 | 7.54e-5 |
| Mix. of Gauss. | **7.64e-2** | 2.50e-4 | 2.88e-5 |
| Mix. of Student T | **1.45e-2** | 2.17e-4 | NA |
| Log. regression | **6.55e-2** | 4.30e-4 | 6.24e-5 |

| | AVS | ANICE | L2HMC | HMC | RWM |
|---|---|---|---|---|---|
| Ring | 35.42 | **807.42** | 6.34 | 624.97 | 688.32 |
| Mix. of Gauss. | 66.08 | **288.10** | 3.24 | NA | NA |
| Mix. of Student T | 74.19 | 45.42 | NA | 1.16 | 6.12 |
| Log. regression | 48.12 | **2660.35** | 87.11 | 180.43 | 1858.15 |

samples for all but the Mixture of Gaussians. Though the more recent neural adaptive samplers have higher effective sample sizes, this comes at a high cost in training time that, in our moderately sized problems, overwhelms their benefit. It was also our experience that training L2HMC is very sensitive to correct tuning of many hyper-parameters. In a search over 42 training configurations only 2 configurations converged to samplers able to mix between modes in the Mixture of Gaussian experiments and none for the Mixture of T. The difficulty of tuning is hard to quantify and will no doubt vary with the experience of the practitioner. We nonetheless see this as a possible barrier to the adoption of more complex adaptive methods and a possible explanation for the popularity of RWM. We found qualitatively that AVS and A-NICE-MC were more straightforwards to tune with good performance from many hyper-parameter configurations. When training time is included, the neural samplers often under-perform HMC but we expect this effect to diminish with data-size.

The work most similar to ours is Variational MCMC (De Freitas et al., 2001), in which a variational approximation is used as a proposal distribution in independent Metropolis-Hastings. To overcome the poor scaling with dimension of the independent Metropolis algorithm, the authors interleave their variational proposal distribution with traditional RWM and make block proposals. Although their use of a variational distribution aids the convergence of MCMC, their method still struggles to mix efficiently even in relatively low dimensional problems. By introducing low dimensional auxiliary variables, we are able both to fit a more accurate approximating distribution and to leverage the learned low dimensional structure to take very large steps in the target space, even hopping between modes. In Salimans et al. (2015) the authors also consider a combination of MCMC and auxiliary variational inference but take a different approach, choosing to use MCMC kernels to design more flexible variational distributions rather than constructing an MCMC sampler at all.

There have also been other recent attempts to parameterize MCMC kernels with neural networks that don't start from variational inference. A-NICE-MCMC (Song et al., 2017), attempts to learn MCMC transition kernels parameterized by volume preserving flows (Dinh et al., 2015). They craft an adversarial objective to train their kernels and use a discriminator that examines pairs of samples, in order to encourage fast mixing. To ensure they have a valid sampler, they use a bootstrap procedure starting from an existing MCMC algorithm. L2HMC (Levy et al., 2018) is an extension of HMC, that parameterizes a scaling and shift of Hamiltonian dynamics with neural networks and trains this parameterization to optimize the expected squared step-size. They show that the introduction of flexible neural networks to HMC vastly improves the ability to mix between modes but still require access to the gradient of the target density.

For any adaptive MCMC method, care has to be taken to preserve the ergodicity of the chain during adaptation (Andrieu & Thoms, 2008; Roberts & Rosenthal, 2009). To overcome this, all of the above methods, including ours, stop adaptation before collecting any samples. A potential problem with this strategy is that if the learned sampler is specialized to only a part of the distribution or the variational approximation has entirely missed regions of high probability, then such regions are unlikely to be explored. A possible avenue for future work would be to investigate iterative improvement of our variational approximation with samples drawn from the true distribution rather than our variational approximation. Another potential avenue for ensuring good coverage of the variational approximation, would be to combine our reverse KL objective, (14), with the KL divergence in the forward direction as this is known to encourage moment matching rather than mode seeking behavior (Minka, 2005).

## 5 CONCLUSION

We introduced a novel framework for combining MCMC and Variational Inference that makes use of the auxiliary variational method to capture low-dimensional latent structure. We have explored

a particular black-box instance of this framework and demonstrated that it can be used to create a fast mixing sampler without the need to take gradients of the target distribution. The method is competitive with other recent geometry-learning approaches and opens up additional avenues for exploring how to combine the best of variational inference and sampling.

## 6 ACKNOWLEDGEMENTS

We would like to thank Harshil Shah, James Townsend and Hippolyt Ritter for useful comments and feedback. This work was supported by the Alan Turing Institute under the EPSRC grant EP/N510129/1.

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

SUPPLEMENTARY MATERIAL

# 7   METROPOLIS-HASTINGS WITH A MIXTURE PROPOSAL

We define a mixture proposal using an auxiliary variable $a$ as

$$\tilde{q}(x'|x) = \int \tilde{q}(x'|a)\tilde{q}(a|x)da \tag{16}$$

We consider the transition kernel

$$q(x', a|x) = \tilde{q}(x', a|x)f(x', a, x) + \delta(x', x)\,\tilde{q}(a|x)\left(1 - \int \tilde{q}(x'', a'|x)f(x'', a', x)dx''da'\right) \tag{17}$$

It is trivial to check that this defines a valid distribution $q(x', a|x)$. We wish to set $f(x', a, x)$ such that $p(x)$ is a stationary distribution of $q(x'|x)$. That is

$$p(x') = \int q(x', a|x)p(x)dxda$$

The right hand side of the above equation can be written as

$$\int \tilde{q}(x', a|x)f(x', a, x)p(x)dxda$$

$$+ \int \delta(x', x)\,\tilde{q}(a|x)\left(1 - \int \tilde{q}(x'', a'|x)f(x'', a', x)dx''da'\right)p(x)dxda \tag{18}$$

which simplifies to

$$\int \tilde{q}(x', a|x)f(x', a, x)p(x)dxda \quad + \quad p(x')\left(1 - \int \tilde{q}(x'', a'|x')f(x'', a', x')dx''da'\right) \tag{19}$$

For the above to hold we require (changing the integration variable $x''$ to $x$ and $a'$ to $a$)

$$\int \tilde{q}(x', a|x)f(x', a, x)p(x)dxda = \int \tilde{q}(x, a|x')f(x, a, x')p(x')dxda \tag{20}$$

Writing

$$\tilde{q}(x', a|x) = \tilde{q}(x'|a)\tilde{q}(a|x) \tag{21}$$

and considering the function

$$f(x', a, x) = \min\left(1, \frac{\tilde{q}(x|a)\tilde{q}(a|x')p(x')}{\tilde{q}(x'|a)\tilde{q}(a|x)p(x)}\right) \tag{22}$$

one can readily verify that

$$f(x', a, x)\tilde{q}(x'|a)\tilde{q}(a|x)p(x) = f(x, a, x')\tilde{q}(x|a)\tilde{q}(a|x')p(x') \tag{23}$$

meaning that (20) is satisfied.

We can then sample from $q(x', a|x)$ by first sampling $a$ from $\tilde{q}(a|x)$ and then sampling from $\tilde{q}(x'|a)$ and accepting with probability $f(x', a, x)$. That is :

1. Sample $a$ from $\tilde{q}(a|x)$
2. Sample $x'$ from $\tilde{q}(x'|a)$
3. Accept the candidate sample $x'$ with probability

$$\min\left(1, \frac{\tilde{q}(x|a)\tilde{q}(a|x')p(x')}{\tilde{q}(x'|a)\tilde{q}(a|x)p(x)}\right) \tag{24}$$

   otherwise reject $x'$ and define the new sample as a copy of the current $x$, namely $x' = x$.

The extension to the case of more than 1-auxiliary variable follows naturally using the same argument as above.

If we were to perform Metropolis-Hastings in the joint space of $(x, a)$ then the acceptance probability would be given by:

$$\min\left(1, \frac{p(a'|x')p(x')\tilde{q}(a, x|a', x')}{p(a|x)p(x)\tilde{q}(x', a'|a, x)}\right)$$

which reduces to (24) if $p(a|x)$ is a constant.

## 8 Exact Parameterization of the variational distributions

### 8.1 Low dimensional examples

For the mixtures of Gaussians and ring density experiments we used an auxiliary dimension of 1 and the target dimension was 2. For the mixture of Gaussians, the variational distributions had the following form:

$$q(a) = \mathcal{N}(a|0, I) \tag{25}$$

$$q_\phi(x|a) = \mathcal{N}(x|\mu_\phi(a), \Sigma_\phi(a)) \tag{26}$$

$$p_\theta(a|x) = \mathcal{N}(a|\mu_\theta(x), \Sigma_\theta(x)) \tag{27}$$

Where the $\mu_\phi(x)$ and $\Sigma_\phi(a)$ are 3 layer feed-forward neural networks with 10 neurons in each layer, all but the last layer were shared between them. Similarly $\mu_\theta(x)$ and $\Sigma_\theta(x)$ were also 3 layer feed-forward neural networks with all but the last layer shared. We used tanh non-linearities in all but the final layer which was simply linear.

In the case of the ring density experiments $p_\theta(a|x) = \Sigma_k \pi_k \mathcal{N}\left(a|\mu_\theta^k(x), \Sigma_\theta^k(x)\right)$ was a mixture of 2 Gaussians, each parameterized as above. The mixture weights $\pi_k$ were also learned and were paramterized as $\pi_1 = \sigma(\tau)$, $\pi_2 = 1 - \pi_1$ to allow for unconstrained optimization.

### 8.2 Regression examples

For Bayesian logistic regression, the target dimension was 14, the auxiliary dimension used was 2 dimensional and the number of hidden units was 300. The structure was otherwise the same as for the low dimensional experiments.

## 9 Calculation of the effective sample size

The effective sample size is calculated as the reciprocal of the auto-correlation-time. It is intended to represent the number of truly independent samples that would be equivalent to a correlated sample drawn using MCMC, in terms of the variance of estimated quantities. One definition of the auto-correlation-time, for a 1-dimensional chain, is:

$$\rho = 1 + 2 \sum_{\tau=1}^{\infty} C_\tau \tag{28}$$

where $C_\tau = E\left[x_t, x_{t+\tau}\right]$ is the lag-$\tau$ auto-correlation of the stationary converged MCMC chain. Though multivariate definitions of the auto-correlation exist, it is common practice to report the lowest effective-sample-size across all dimensions. For ease of comparison, we adopt this practice.

There are numerous methods for estimating $\rho$ (Thompson, 2010). It is worth noting however that simply estimating $C_\tau$ from multiple chains and substituting the estimates into the above formula does not yield a consistent estimator except in very simple cases when the auto-correlations are guaranteed to be positive. In general however, the auto-correlations can be of both negative and positive variance. The estimator formed by substituting the empirically estimated auto-correlations does not have a variance that goes to 0 as the sequence length goes to infinityThompson (2010). Instead we use the batch-means estimator.

### 9.1 batch-means estimator

An estimator of the auto-correlation time is computed using "batch-means" (Thompson, 2010). A single long sequence is split into $m$ sub-sequences and the mean of each sub-sequence is calculated. The auto-correlation-time is then estimated by using the ratio of the variance of the batch-means to the variance of the overall sequence. The estimator is given by:

$$\hat{\rho} = m \frac{s_m^2}{s^2} \tag{29}$$

where $s_m^2$ is the variance of the batch-means and $s^2$ is the variance of the entire chain.

## 10 FURTHER EXPERIMENTAL DETAILS

### 10.1 ABSOLUTE TRAINING TIMES

Table 3: Absolute training times in seconds for each of the learned samplers for the experiments given in section 3, rounded to the nearest second.

|                   | AVS | ANICE | L2HMC |
|-------------------|-----|-------|-------|
| Ring              | 70  | 120   | 3310  |
| Mix. of Gauss.    | 2   | 1289  | 2619  |
| Mix. of Student T.| 3   | 290   | NA    |
| Log. regression   | 2   | 598   | 4590  |

