# OpenReview forum: "Auxiliary Variational MCMC"
_ICLR.cc/2019/Conference_

### Official Review · AnonReviewer3 · 2018-11-02
**Interesting paper, though results are slightly weak.**

**Rating:** 7
**Confidence:** 4

**Review:**

This paper proposes an auxiliary variable MCMC scheme involving variational inference for efficient MCMC. Given a target distribution p(x), the authors introduce an auxiliary variable a, and learn conditional distributions p(a|x) and q(a|x) by minimizing the KL divergence between p(x)p(a|x) and q(a)q(x|a), with q(a) something simple (the authors use Gaussian). A MH proposal step involves simulating x givea the current MCMC sample x (from p(a|x), taking a step in A-space, and then returning back to the X space (using q(x|a)).  The authors show how to calculate the acceptance probability.

I think the idea is nice and useful (I'm surprised people haven't thought of this before), though I think the paper presents this in a less clear way (as an extension of ideas from Agakov and Barber's "Auxiliary variational method"). While this is correct and perhaps more general, in my mind it slightly obscures the main idea, as well as the strong ties with variational autoencoders: express a complex distribution as a (learnt) transformation of a simple distribution (this is the actual approach taken in the experiments).

The motivation of the approach is that the nonlinear encoding network can transform the complex p(x) into a simpler q(a).
For this reason, I think an important baseline is the independent MH sampler from equation 8 (I think this essentially uses a trained VAE generative model as a proposal distribution). The authors talk about how producing independent proposals can be sub-optimal, yet it seems to me that if the encoder and decoder neural networks are powerful enough, this should do a good job. I think excluding this baseline hurts the paper a bit.

The proof of correctness while correct is a bit unclear, can perhaps be simplified if you view the MCMC algorithm as operating on an augmented space (x,a,x') with stationary distribution p(x)q(a|x)q(x'|a) (writing writing q for \tilde(q)). This clearly has the right distribution over x. Each MCMC iteration starts with x and proceeds as follow:
  1) Given x, sample a and x' from q(a|x) and q(x'|a)
  2) Make a deterministic proposal on the augmented space to swap (x,x'). The acceptance probability is now equation 2.
  3) Discard a,x'.

In figure 4, the authors use HMC as an "improved MCMC algorithm", yet this is not an algorithm that deals with multimodality well. More useful would be to include some tempering algorithm like serial or parallel tempering.

While I like the idea, I unfortunately don't think the experiments are very convincing (and the authors barely discuss their results). Other than mixture of Gaussians, HMC (which involves no training) appears to be superior. With some tempering, I expect it to outperform the proposed method for the MoG case

Table 2 left: since HMC involves no training, does this mean that, taking training time into account, HMC is 5-6 orders of magnitude more efficient. L?ke I mentioned earlier, these results need more discussion.

It would also help to provide absolute training and run times, so the reader can better understand whether the proposed method of ANICE is better.

Figure 3: why don't the authors also plot the histogram of values in the auxiliary space, p(a). It would be interesting to see how Gaussian this is (this is what variational inference is trying to achieve). Also, does Figure 3(a) mean that conditioned on x, p(a|x) is basically a delta function? This would suggest that the encoder is basically learning a deterministic transformation to a simpler low-dimensional space? There is some work in this direction in the statistics literature, e.g.
"Variable transformation to obtain geometric ergodicity in the random-walk Metropolis algorithm"

The authors some refers to the distribution of a|x as q(a|x) sometimes (in section 2.1) and sometimes as p(a|x) which is a bit confusing.

Figure 2: the labels are wrong.

---

> ### Author Response · Authors · 2018-11-16
> **Thanks and Response - Part 2 of 2**
>
>
> >>While I like the idea, I unfortunately don't think the experiments are very convincing (and the authors barely discuss their results). Other than mixture of Gaussians, HMC (which involves no training) appears to be superior. With some tempering, I expect it to outperform the proposed method for the MoG case
>
> >>Table 2 left: since HMC involves no training, does this mean that, taking training time into account, HMC is 5-6 orders of magnitude more efficient. Like I mentioned earlier, these results need more discussion.
>
>
> >>It would also help to provide absolute training and run times, so the reader can better understand whether the proposed method of ANICE is better.
>
> The reviewer is correct that, presently, if training time is included HMC significantly outperforms not just the method proposed in this paper but all of the learned samplers explored thus far. The size of the difference is dramatic here because the data-sets used are small but as the size of the data-set grows we expect the computational cost of HMC to grow much faster than the learned samplers. This is because HMC will require many gradients of the log-density per iteration and the cost of each of these scales linearly in the size of the data. Whilst calculating acceptance ratios also scales linearly in data-size, it only needs to be performed once per iteration. HMC, on the other hand, will often require 10s or 100s of gradients per iteration. Another point that we'd emphasise is that although there is an increased computational time for these methods, because they are black-box they potentially save on much more vital human time, HMC being quite tricky to tune.
>
> We will update the paper to include absolute training times. To give the reviewer an indication here of the difference in absolute training times (in seconds) we provide preliminary figures taken from the last batch of experiments we ran:
>
> Ring/   Mog/   Logistic Regression/
>
> ANICE:  120/ 1290 / 3310
>
> AVS: 70 / 2  /   2
>
> L2HMC: 3310/ 2620 / 4590
>
> We will try to expand the discussion of the results, their brevity was primarily a result of space restrictions.
>
> >>Figure 3: why don't the authors also plot the histogram of values in the auxiliary space, p(a). It would be interesting to see how Gaussian this is (this is what variational inference is trying to achieve).
>
> We agree that this might be interesting but felt that in the limited space given it wasn't central to our argument.
>
> >> Also, does Figure 3(a) mean that conditioned on x, p(a|x) is basically a delta function? This would suggest that the encoder is basically learning a deterministic transformation to a simpler low-dimensional space? There is some work in this direction in the statistics literature, e.g.
>
> In this case, yes the conditional became close to a delta-function though its not clear how often this will be the case.
>
> >>"Variable transformation to obtain geometric ergodicity in the random-walk Metropolis algorithm"
>
> Thanks also for the pointer to this paper. We were not aware of it but it seems highly relevant. We'll try to add it to our discussion after we study it in more detail.
>
> >>The authors some refers to the distribution of a|x as q(a|x) sometimes (in section 2.1) and sometimes as p(a|x) which is a bit confusing.
>
> Thanks this was a mistake and will be corrected.
>
> >>Figure 2: the labels are wrong.
>
> Thanks again, we will correct this.

---

> ### Author Response · Authors · 2018-11-16
> **Thanks and Response - Part 1 of 2**
>
> Thank you for your review and well considered comments.
>
> >>This paper proposes an auxiliary variable MCMC scheme involving variational inference for efficient MCMC. Given a target distribution p(x), the authors introduce an auxiliary variable a, and learn conditional distributions p(a|x) and q(a|x) by minimizing the KL divergence between p(x)p(a|x) and q(a)q(x|a), with q(a) something simple (the authors use Gaussian). A MH proposal step involves simulating x givea the current MCMC sample x (from p(a|x), taking a step in A-space, and then returning back to the X space (using q(x|a)).  The authors show how to calculate the acceptance probability.
>
> >>I think the idea is nice and useful (I'm surprised people haven't thought of this before), though I think the paper presents this in a less clear way (as an extension of ideas from Agakov and Barber's "Auxiliary variational method"). While this is correct and perhaps more general, in my mind it slightly obscures the main idea, as well as the strong ties with variational autoencoders: express a complex distribution as a (learnt) transformation of a simple distribution (this is the actual approach taken in the experiments).
>
> We did consider presenting the exposition in this way but in the end decided to err on the side of generality. Ultimately, we think that there are really two main ideas in the paper: 1) A quite general framework for construction of MH proposals that can exploit structure 2) A VAE-inspired black-box instantiation of that structure.
>
> Though we primarily investigated the neural-net version of our sampler, we do think that there are likely other ways to construct valid samplers that may be more efficient on a problem specific basis and wanted to expose this possibility to the research community.
>
>
> >>The motivation of the approach is that the nonlinear encoding network can transform the complex p(x) into a simpler q(a).
> >>For this reason, I think an important baseline is the independent MH sampler from equation 8 (I think this essentially uses a trained VAE generative model as a proposal distribution). The authors talk about how producing independent proposals can be sub-optimal, yet it seems to me that if the encoder and decoder neural networks are powerful enough, this should do a good job. I think excluding this baseline hurts the paper a bit.
>
> We think this is a fair point and will run this experiment and update the results.
>
> >>The proof of correctness while correct is a bit unclear, can perhaps be simplified if you view the MCMC algorithm as operating on an augmented space (x,a,x') with stationary distribution p(x)q(a|x)q(x'|a) (writing writing q for \tilde(q)). This clearly has the right distribution over x. Each MCMC iteration starts with x and proceeds as follow:
> >>  1) Given x, sample a and x' from q(a|x) and q(x'|a)
> >>  2) Make a deterministic proposal on the augmented space to swap (x,x'). The acceptance probability is now equation 2.
> >>  3) Discard a,x'.
>
> This point was also made by another reviewer who suggested a slightly different approach that was also valid. As we said to them, we agree that there are perhaps more direct proofs of our method. However, rigorously handling deterministic proposals in Metropolis Hastings is quite an advanced topic and we feel that our proof whilst algebraically more involved is conceptually simpler. Unless the reviewers feel very strongly on this point, the authors would prefer to maintain the proof as is.
>
>
> >>In figure 4, the authors use HMC as an "improved MCMC algorithm", yet this is not an algorithm that deals with multimodality well. More useful would be to include some tempering algorithm like serial or parallel tempering.
>
> In retrospect the inclusion of HMC here is maybe a little distracting. The point was not to demonstrate superiority over advanced methods but simply to demonstrate the ability of our sampler to find low dimensional structure when we know it is present. Perhaps the main reason we chose to include this example though, was because the inability of HMC to mix between modes was a key problem investigated in the L2HMC paper, which we benchmark against. The authors would happily remove this example if the reviewer feels it adds little.

---

### Official Review · AnonReviewer1 · 2018-11-02
**The paper contains very interesting novel ideas. Some points must clarified and the state-of-the-art must be improved.**

**Rating:** 7
**Confidence:** 4

**Review:**

In my opinion, the paper contains very interesting novel ideas.
However, some parts needs a future clarification and the state-of-the-art must be improved.

- First of all,  Sections 2.3.1 or 2.3.2 can be improved and clarified. For instance, I believe you can create a unique section with title " Choice of Proposal density " and then schematically describe each proposal from the simplest to the more sophisticated one.

- At the beginning of Section 2, please devote more sentence to explain why extending the space and apply the variational inference is good for finding a suitable good proposal density.

- Related  to Section 2 ( theMixture Proposal MCMC contribution), the authors should discuss (in the introduction and also in the related works section) the Multiple Try Metropolis schemes with correlated candidates where, for instance, a path of candidates is generated and one of them is selected and tested with MH-type acceptance probability, in a proper way. This is more general that your scheme but very related. Please see

Qin, Z.S., Liu, J.S., 2001. Multi-point Metropolis method with application to hybrid Monte Carlo. Journal of Computational Physics 172, 827–840.

L. Martino, V. P. Del Olmo, J. Read, "A multi-point Metropolis scheme with generic weight functions", Statistics and Probability Letters, Volume 82, Issue 7, Pages: 1445-1453, 2012.

L. Martino, "A Review of Multiple Try MCMC algorithms for Signal Processing", Digital Signal Processing, Volume 75, Pages: 134-152, 2018.

- Related again with the state-of-the-art description, the references regarding  Adaptive Mixture Metropolis methods are completely missed. If I have properly understood, you also adapt a mixture via variational inference. Please, in Section 4, consider the different works that considers an adapting mixture proposal for a  Metropolis-type algorithm,

P. Giordani and R. Kohn, “Adaptive independent Metropolis-Hastings by fast estimation of mixtures of normals,” Journal of Computational and Graphical Statistics, vol. 19, no. 2, pp. 243–259, September 2010.

Tran, M.-N., M. K. Pitt, and R. Kohn. Adaptive Metropolis–Hastings sampling using reversible dependent mixture proposals. Statistics and Computing, 26, 1–21, 2014.

D. Luengo, L. Martino, "Fully Adaptive Gaussian Mixture Metropolis-Hastings Algorithm", IEEE International Conference on Acoustics, Speech, and Signal Processing (ICASSP), Vancouver (Canada), 2013.

Roberts, G. O. and J. S. Rosenthal (2009). Examples of adaptive MCMC. Journal of Computational and Graphical Statistics 18, 349–367.

---

> ### Author Response · Authors · 2018-11-16
> **Thanks and Response**
>
> Thank you for your review and well considered comments.
>
> >> Review: In my opinion, the paper contains very interesting novel ideas.
> >> However, some parts needs a future clarification and the state-of-the-art must be improved.
>
>
> >> First of all,  Sections 2.3.1 or 2.3.2 can be improved and clarified. For instance, I believe you can create a unique section with title " Choice of Proposal density " and then schematically describe each proposal from the simplest to the more sophisticated one.
>
> Thanks for the feedback. As we edit the paper to include other changes we'll bear this in mind. We'd hoped this was what we had done already but will try to make it clearer.
>
> >> At the beginning of Section 2, please devote more sentence to explain why extending the space and apply the variational inference is good for finding a suitable good proposal density.
>
> We believe that this was made clear in sections 2.3.1 and 2.3.2. In particular the discussion immediately following equation (9) tries to make this point. However, we can add a further sentence emphasising this at the start of section 2 as well.
>
> >> Related  to Section 2 ( theMixture Proposal MCMC contribution), the authors should discuss (in the introduction and also in the related works section) the Multiple Try Metropolis schemes with correlated candidates were, for instance, a path of candidates is generated and one of them is selected and tested with MH-type acceptance probability, in a proper way. This is more general that your scheme but very related. Please see
>
> >> Qin, Z.S., Liu, J.S., 2001. Multi-point Metropolis method with application to hybrid Monte Carlo. Journal of Computational Physics 172, 827–840.
>
> >> L. Martino, V. P. Del Olmo, J. Read, "A multi-point Metropolis scheme with generic weight functions", Statistics and Probability Letters, Volume 82, Issue 7, Pages: 1445-1453, 2012.
>
> >>L. Martino, "A Review of Multiple Try MCMC algorithms for Signal Processing", Digital Signal Processing, Volume 75, Pages: 134-152, 2018.
>
> Thanks for the pointers to these papers. We are aware of Multiple Try Metropolis (MTM) but many of the references you provided below were new to us. Whilst we acknowledge that MTM is a powerful tool in the MCMC arsenal, we felt that it was quite different to our method and really offers an orthogonal direction for improvement. We don't attempt a thorough review of the state-of-the-art in MCMC, which we feel is beyond the scope here, but instead try to focus our discussion on other neural adaptive samplers such as L2HMC and A-NICE-MCMC.
>
>
> >> related again with the state-of-the-art description, the references regarding  Adaptive Mixture Metropolis methods are completely missed. If I have properly understood, you also adapt a mixture via variational inference. Please, in Section 4, consider the different works that considers an adapting mixture proposal for a  Metropolis-type algorithm,
>
> Thanks for the pointers to these papers. These were mostly new to us and do seem very related. After reading the papers more closely we will try to include them in our references.
>
> >>P. Giordani and R. Kohn, “Adaptive independent Metropolis-Hastings by fast estimation of mixtures of normals,” Journal of Computational and Graphical Statistics, vol. 19, no. 2, pp. 243–259, September 2010.
>
> >>Tran, M.-N., M. K. Pitt, and R. Kohn. Adaptive Metropolis–Hastings sampling using reversible dependent mixture proposals. Statistics and Computing, 26, 1–21, 2014.
>
> >>D. Luengo, L. Martino, "Fully Adaptive Gaussian Mixture Metropolis-Hastings Algorithm", IEEE International Conference on Acoustics, Speech, and Signal Processing (ICASSP), Vancouver (Canada), 2013.
>
> >>Roberts, G. O. and J. S. Rosenthal (2009). Examples of adaptive MCMC. Journal of Computational and Graphical Statistics 18, 349–367

---

### Official Review · AnonReviewer4 · 2018-11-08
**A very interesting idea for combining MCMC and VI.**

**Rating:** 7
**Confidence:** 5

**Review:**

This paper proposes a clever and sensible approach to using the structure learned by the auxiliary variational method to accelerate random-walk MCMC. The idea is to learn a low-dimensional latent space that explains much of the variation in the original parameter space, then do random-walk sampling in that space (while also updating a state variable in the original state, which is necessary to ensure correctness).

I like this idea and think the paper merits acceptance, although there are some important unanswered questions. For example:
- How does the method work on higher-dimensional target distributions? I would think it would be hard for a low-dimensional auxiliary space to have high mutual information with a much higher-dimensional space. In principle neural networks can do all sorts of crazy things, but phenomena like VAEs with low-dimensional latent spaces generating blurry samples make me suspect that auxiliary dimension should be important.
- How does the method work with hierarchical models, heavy-tailed models, etc.? Rings, MoGs, and flat logistic regressions are already pretty easy targets.
- Is it really so valuable to not need gradients? High-quality automatic differentiation systems are widely available, and variational inference on discrete parameters with neural nets remains a pretty hard problem in general.

Some other comments:

* It’s probably worth citing Ranganath et al. (2015; “Hierarchical Variational Models”), who combine the auxiliary variational method with modern stochastic VI. Also, I wonder if there are connections to approximate Bayesian computation (ABC).

* I think you could prove the validity of the procedure in section 2.1 more succinctly by interpreting it as alternating a Gibbs sampling update for “a” with a Metropolis-Hastings update for “x”. If we treat “a” as an auxiliary variable such that
p(a | x) = \tilde q(a | x)
p(x | a) \propto p(x) \tilde q(a | x)
then the equation (2) is the correct M-H acceptance probability for the proposal
\tilde q(a’, x’) = δ(a’-a) \tilde q(x’ | a).
Alternating between this proposal and a Gibbs update for “a” yields the mixture proposal in section 2.1.

* It’s also possibly worth noting that this procedure will have a strictly lower acceptance rate than the ideal procedure of using the marginal
\tilde q(x’|x)
as a M-H proposal directly. Unfortunately that marginal density usually can’t be computed, which makes this ideal procedure impractical. It might be interesting to try to say something about how large this gap is for the proposed method.

* "We choose not to investigate burn-in since AVS is initialized by the variational distribution and therefore has negligible if any burn-in time.” This claim seems unjustified to me. It’s only true insofar as the variational distribution is an excellent approximation to the posterior (in which case why use MCMC at all?). It’s easy to find examples where an MCMC chain initialized with a sample from a variational distribution takes quite a while to burn in.

---

> ### Author Response · Authors · 2018-11-16
> **Thanks and Response - Part 2 of 2**
>
> >> * I think you could prove the validity of the procedure in section 2.1 more succinctly by interpreting it as alternating a Gibbs sampling update for “a” with a Metropolis-Hastings update for “x”. If we treat “a” as an auxiliary variable such that
>
> >>p(a | x) = \tilde q(a | x)
> >>p(x | a) \propto p(x) \tilde q(a | x)
> >>then the equation (2) is the correct M-H acceptance probability for the proposal
> >>\tilde q(a’, x’) = δ(a’-a) \tilde q(x’ | a).
> >>Alternating between this proposal and a Gibbs update for “a” yields the mixture proposal in section 2.1.
>
> We agree that there are perhaps more direct proofs of our method, as has been suggested by another reviewer as well. However, rigorously handling deterministic proposals in Metropolis Hastings is quite an advanced topic and we feel that our proof whilst algebraically more involved is conceptually simpler. We also feel that the extension of the proof as given to multiple auxiliary variables is more straight forward. Unless the reviewers feel very strongly on this point, we would prefer to maintain the proof as is.
>
> >> It’s also possibly worth noting that this procedure will have a strictly lower acceptance rate than the ideal procedure of using the marginal
> >>\tilde q(x’|x)
> >>as a M-H proposal directly. Unfortunately that marginal density usually can’t be computed, which makes this ideal procedure impractical. It might be interesting to try to say something about how large this gap is for the proposed method.
>
> This is an interesting question and not one we had investigated in detail. We shall think carefully about this and if we have any insights prior to the final deadline shall update the paper.
>
> >> "We choose not to investigate burn-in since AVS is initialized by the variational distribution and therefore has negligible if any burn-in time.” This claim seems unjustified to me. It’s only true insofar as the variational distribution is an excellent approximation to the posterior (in which case why use MCMC at all?). It’s easy to find examples where an MCMC chain initialized with a sample from a variational distribution takes quite a while to burn in.
>
> Perhaps the claim as stated is slightly too strong but it seems plausible to the authors that initialising with a variational distribution will likely start the chain in a region of high target density even if the variational approximation is poor. This in turn will reduce burn-in time relative to a method that is not initialised in a region of high density. Indeed, this is the reason for the common practice of initialising MCMC algorithms by first running an optimisation procedure to find the mode. In any case, all the authors were hoping to convey was that burn-in is relatively fast for AVS and thus not the most interesting point of comparison relative to other methods.
>
> We will amend the text to read: "We choose not to focus our investigation on burn-in time since AVS, being initialized by the variational distribution, often has short burn-in relative to competitive methods.”

---

> ### Author Response · Authors · 2018-11-16
> **Thanks and Response - Part 1 of 2**
>
> Thank you for your review and well considered comments.
>
> >> Review: This paper proposes a clever and sensible approach to using the structure learned by the auxiliary variational method to accelerate random-walk MCMC. The idea is to learn a low-dimensional latent space that explains much of the variation in the original parameter space, then do random-walk sampling in that space (while also updating a state variable in the original state, which is necessary to ensure correctness).
>
> >> I like this idea and think the paper merits acceptance, although there are some important unanswered questions. For example:
>
>
> >> How does the method work on higher-dimensional target distributions? I would think it would be hard for a low-dimensional auxiliary space to have high mutual information with a much higher-dimensional space. In principle neural networks can do all sorts of crazy things, but phenomena like VAEs with low-dimensional latent spaces generating blurry samples make me suspect that auxiliary dimension should be important.
>
> This is an interesting question and one we think it is hard to give a definitive answer to. The degree to which we can benefit from learning low dimensional structure will depend significantly on the choice of target distribution. It is easy to construct very high dimensional distributions with low dimensional parametrisations and in these cases it is likely the method will work well but it's unclear how many of the posteriors typically encountered in real world Bayesian inference actually have this structure. In practice, we have found the method to perform reasonably well on problems up till 10's of dimensions, which would cover quite a wide range of statistical applications. However, we were not able to get our method (or any of the other neural adaptive samplers tested) to sample reliably from the posterior of a relatively small neural network.
>
>  We would argue that the blurry images often produced by VAEs are more to do with the training objective than the ability of neural nets to discover low-dimensional structure. This is evidenced by the fact that otherwise identical models, when trained with an adversarial objective are able to produce very sharp images.
>
> >>  How does the method work with hierarchical models, heavy-tailed models, etc.? Rings, MoGs, and flat logistic regressions are already pretty easy targets.
>
> This is a fair point and we will try to add one or two more experiments and update the paper.
>
> >> Is it really so valuable to not need gradients? High-quality automatic differentiation systems are widely available, and variational inference on discrete parameters with neural nets remains a pretty hard problem in general.
>
> We take your point but would still argue that the ability to avoid gradient computations could be of benefit, especially in the large data regime.
>
> Though automatic differentiation eases the implementation burden, it doesn't do away with the computational difficulty of calculating gradients which for most Bayesian inference will scale linearly in the size of the data-set and for exact HMC will need to be calculated multiple times per iteration (sometimes even hundreds oft times). It's exactly for this reason that methods such as Stochastic Gradient HMC have been introduced but these methods are derived for continuous time and are not exact in practice.
>
> Furthermore, whilst it's true that discrete neural variational inference (NVI) remains challenging it is an area of very active ongoing research and this method opens up the possibility to translate progress in NVI immediately to MCMC.
>
> >> Some other comments:
>
> >>* It’s probably worth citing Ranganath et al. (2015; “Hierarchical Variational Models”), who combine the auxiliary variational method with modern stochastic VI. Also, I wonder if there are connections to approximate Bayesian computation (ABC).
>
> Thanks for this pointer, we were not aware of this paper and it does indeed use many of the same ingredients. We will add this reference.

---

### Public Comment · (anonymous) · 2018-11-13
**Gradient use during training phase of AVS**

In the training phase of AVS, we use the reparameterization gradient estimator which requires the gradient of the target distribution (e.g., p wrt to x). Is that understanding correct?

---

> ### Author Response · Authors · 2018-11-13
> **Gradients only with respect to the variational parameters**
>
> Hi,
>
> thanks for your interest in our paper.  Your suggestion is not quite right.
>
> We want to optimise a KL objective,  KL[p_\theta(a|x) p(x) | q_\phi(x|a) q(a)],  with respect to the parameters of the variational distributions. We are free here to choose the forms of the variational distribution and so have a significant amount of control. The gradients are not taken with respect to the variable x and so the gradient of the target density p(x) is never required even during training.
>
> The re-parameterisation gradient becomes useful when we set q_\phi(x|a) = N(\mu(a), \sigma^2(a)). Here we have set q as a Gaussian whose mean and variance are parameterised by neural networks. We use the re-parameterisation trick to take gradients of expectations with respect to this distribution. In particular we use it to get low variance gradients of the KL stated above.
>
> Hope that clears things up but please feel free to follow up.

---

> > ### Public Comment · (anonymous) · 2018-11-13
> > **Are you sure?**
> >
> > The paper claims to use the reverse KL. Is that a typo?
> >
> > Computing the gradient of KL[ q_\phi(x|a) q(a) || p_\theta(a|x) p(x)] wrt to the parameters of q_\phi(x | a) using the reparameterization trick requires the gradient of p(x) wrt to x. This is done implicitly by TensorFlow when you take the gradient with a reparameterized x. Are you sure this isn't happening in your code?

---

> > > ### Author Response · Authors · 2018-11-15
> > > **You are correct**
> > >
> > > Thanks again for your comment.
> > >
> > > Yes we do use the reverse KL, that was indeed a typo in the comment.
> > >
> > > Apologies, I didn't fully appreciate your question before. You are right that when we re-parametrise x = \mu_theta + \sigma_theta * \epsilon, there will be an implicit gradient of the target density required during training. This means that presently in the paper we do use the gradient of the log-density during training, though not when sampling. It's worth noting though that it's not strictly necessary to re-parameterise. If the gradient cost was prohibitive, we could take the stochastic gradient, without calculating derivatives of p(x), by using the log-derivative trick. Thank you for highlighting this. We'll amend the paper at the end of section 2.4 to clarify this point.
> > >
> > > After training, sampling is unaffected by this observation.

---

> > > > ### Public Comment · (anonymous) · 2018-11-15
> > > > **Thank you for the clarification**
> > > >
> > > > Thank you for the clarification. Agreed that training could be done w/ log-derivative trick, so this doesn't affect the main message of the paper.

---

### Author Response · Authors · 2018-11-25
**Updated Paper**

We would like again to thank the reviewers for their suggested improvements. We have updated the paper to try and improve the exposition in line with their comments. The changes we have made are:

Experiments
1) We include as a baseline the independent proposal distribution (equation (8)) discussed at the start of section (2). We found this proposal to under-perform even our low expectations. The independent proposal failed to sample from all of the test densities apart from the mixture of Gaussians, getting quickly trapped in regions of high-density. For the Mixture of Gaussians we include the ESS results.

2) We have added an experiment for distributions with multi-modality and heavy tails, a mixture of Student-T distributions.*

3) We have included the absolute training times in the appendix to aid comparison of the results.

Exposition

1) We have corrected the minor errors in typography and labelling that we were made aware of.

2) We have removed the distracting comparison to HMC in figure 4.

3) We have slightly expanded the discussion of the results though still find ourselves constrained by the page limit. If the reviewers thought it appropriate we believe we could expand this discussion further but may need to use a 9th page.

4) We have added references to Ranganath et al. (2015; “Hierarchical Variational Models”) and Roberts et al. (2009; Examples of adaptive MCMC. Journal of Computational and Graphical Statistics 18, 349–367).

5) We have softened our statements in regard to burn-in time.

We would be grateful for any further improvements the reviewers might suggest and would happily continue to modify the paper prior to the final deadline.

* At present we haven't been able to reliably draw samples from the mixture of student-T distributions using L2HMC. We think however that this may be a difficulty of parameter tuning and finding an appropriate annealing schedule for the temperature. For now we have omitted this result but if we are able to correctly tune the sampler, we would include the result in the camera ready version of our paper.

---

### Meta-Review · Area_Chair1 · 2018-12-13
**Interesting and novel paper, but experimental results could be more convincing**

**Confidence:** 5
**Recommendation:** Accept (Poster)

**Metareview:**

The reviewers all argued for acceptance citing the novelty and potential of the work as strengths.  They all found the experiments a little underwhelming and asked for more exciting empirical evaluation.  The authors have addressed this somewhat by including multi-modal experiments in the discussion period.  The paper would be more impactful if the authors could demonstrate significant improvements on really challenging problems where MCMC is currently prohibitively expensive, such as improving over HMC for highly parameterized deep neural networks.  Overall, however, this is a very nice paper and warrants acceptance to the conference.